# Renewable Energy: A Curse or Blessing—International Evidence

**Ruoxuan Li [1], Huwei Wen [1,2] , Xinpeng Huang [1] and Yaobin Liu [1,*]**

[1] School of Economics and Management, Nanchang University, Nanchang 330031, China; liquanquanseer@163.com (R.L.); wenhuwei@ncu.edu.cn (H.W.); nse_hxp@foxmail.com (X.H.)

[2] Research Center of the Central China for Economic and Social Development, Nanchang University, Nanchang 330031, China

[*] Correspondence: liuyaobin@ncu.edu.cn

**Abstract:** The development of renewable energy has effectively promoted the process of reaching global carbon neutrality. However, the academic community has not reached a consensus on whether the development of renewable energy will inhibit economic growth. The crux of the debate centers around whether renewable energy paradigms ignore differences in the structure of factor endowments across countries. The panel data of 125 countries from 1990 to 2021 were used to perform group regression for countries with different factor endowment structures. The results show that the renewable energy curse of developed countries becomes stronger and weaker with economic development; the renewable energy curse in developing countries is growing with economic growth; and the economic development of countries with poor natural resources is more vulnerable to the negative impact of renewable energy development. The group regression results of different development stages of renewable energy show that the negative impact of renewable energy development on economic development is not significant in the early stage, but that it has significant impacts in the growth and maturity stage. The mechanism test found that the development of renewable energy affected changes in trade structure and inhibited economic growth.

**Keywords:** renewable energy; low-carbon development; renewable energy policy; energy transition; economic growth; international evidence

## 1. Introduction

The World Meteorological Organization warned the 27th Conference of the Parties to the United Nations Framework Convention on Climate Change (COP27) (https://cop27.eg/assets/files/days/COP27%20INNOVATIVE%20FINANCE-DOC-01-EGY-10-22-EN.pdf, accessed on 13 December 2022) climate conference that greenhouse gas emissions have reached record high levels over the past eight years. According to United Nations Environment Program's 2022 Emissions Gap Report, global temperature will rise by 2.4 to 2.6 °C by the end of the century under current emission reduction policies. In order to jointly address the challenge of climate change and reduce the devastating impact of climate change on human well-being and natural systems, government action is especially urgent in the context of global carbon neutrality. The development of renewable energy is the key to solving the global climate problem and the contradiction between energy supply and demand [1]. Countries around the world have put forward renewable energy policies to promote the development of renewable energy, reverse the fossil-based energy structure, and point the transformation of energy structure in a low-carbon, carbon-free, and clean direction. Countries around the world are using less fossil energy and the consumption of renewable energy is rising significantly, and it is reasonable to state that the development of renewable energy is now the general trend [2]. According to the International Energy Agency, renewables will account for more than 90 per cent of global electricity expansion within five years. This means that renewable energy will develop at a faster speed and play a pivotal role in the energy structure. Technological innovation underpins the process

of developing renewable energy and drives economic development [3]. The scale effect and learning effect of renewable energy reduce the marginal cost and bring more economic benefits [4]. At the same time, the development of renewable energy reduces the economic problems caused by fluctuation in the price of fossil energy. However, renewable energy may also come at an economic cost [5]. Firstly, when considering both economic and ecological benefits, the renewable energy industry may give priority to ecological benefits and sacrifice economic interests. Secondly, the high cost of renewable energy research and development squeezes the output of the government and the private sector: various policies supporting renewable energy development by the government reduce other government expenditures under a certain government income. Furthermore, private sector investment and consumption may be reduced by high prices and taxes for renewable energy products borne by downstream enterprises or consumers. Whether the development of renewable energy is a blessing or a curse, whether there is a curse of renewable energy, and how to correctly implement renewable energy policy to achieve economic and ecological benefits have become issues worthy of further study for policy makers.

The sample selection of existing research is confined to local areas, and the conclusions obtained are special rather than of general guiding significance. This paper uses the cross-country panel data of 125 countries from 1990 to 2021 to examine the impact of renewable energy development on economic development. Figure 1 roughly simulates the relationship between the two. It mainly solves the following questions: first, will the development of renewable energy globally be at the cost of economic development? Second, will there be differences in the impact of renewable energy on economic development in different development stages? Third, will there be differences in the impact of renewable energy development on economic development among countries with different endowment structures? This paper uses the panel quantile model to identify the evolutionary trajectory of the marginal effects of economic growth at different stages of renewable energy development. Grouped regression is performed according to different endowment conditions in the region. After the robustness test using the shock IV method, replacing variables and controlling macro factors, the interaction term is introduced to discuss the mechanism of renewable energy development affecting economic development. The empirical results show that: (1) the development of renewable energy has a negative inhibitory effect on economic development in the world as a whole. (2) The negative impact of renewable energy on economic development in the early stage is not significant. As renewable energy enters the growth and maturity stage, the development of renewable energy has a more significant inhibitory effect on economic development. (3) The average impact of renewable energy development on countries and regions with different endowment structures is different.

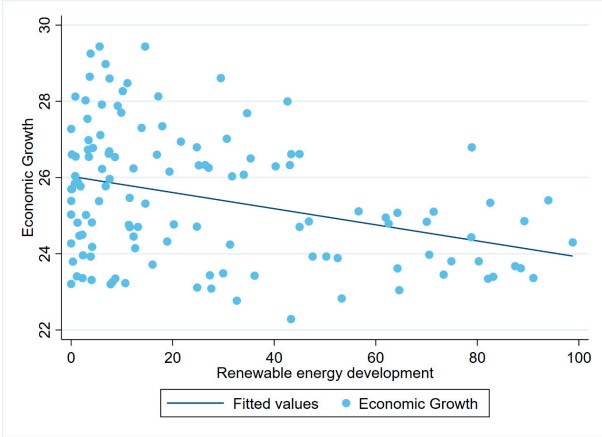

**Figure 1.** The impact of renewable energy development on economic development.

## 2. Review of the Literature and Theoretical Analysis

### 2.1. Review of the Literature

The existing literature on renewable energy mainly focuses on the impact of renewable energy on the environment. It is believed that the development of renewable energy can reduce pollution, significantly improve environmental quality, and alleviate global environmental problems [6]. There is a limited amount of research on the economic impact of renewable energy development. The extant literature uses a variety of indicators, methods, and samples to study the relationship between the development of renewable energy and the wider economy, but no consistent conclusion has been reached. There are primarily three viewpoints: the first is known as the growth hypothesis. Held by a majority of scholars, this belief states that the development of renewable energy promotes the economic development of a country or region. The second, termed the conservation hypothesis, states renewable energy development inhibits the economic development of a country or region. Third, some scholars believe that the impact of renewable energy development on economic development is not significant, and the beliefs of these researchers are termed the neutrality hypothesis.

In terms of empirical evidence for the growth hypothesis, Thuy and Huyen [7], who evaluated the relationship between renewable energy and economic growth in Vietnam from 1995 to 2019, found a long-term positive monocausal relationship between renewable energy use and economic growth. Chiu-Lan Chang and Ming Fang [8] found that renewable energy development has beneficial effects on the economy. Muhammad Mohsin et al. [9] analyzed a sample of 25 Asian economies and concluded that renewable energy contributes to economic development. Yajun Zhang et al. [10] proved through empirical testing that, under the Organization for Economic Cooperation and Development (OECD)'s sample, increased investment in the renewable energy sector has a positive impact on economic output.

In terms of empirical evidence for the conservation hypothesis, Alycia Leonard et al. [11] argued that the resource curse is also present in renewable energy and assessed the risk framework for renewable energy, using Morocco as a case study. Additionally, their research group continues to risk-assess the renewable energy resource curse in low- and middle-income countries [11]. Oluwatobi et al. [12], who looked at the Nigerian economy for 60 years, concluded that in the long run, increased use of renewable energy would reduce GDP growth. Rehman et al. [13] used panel data of the Group of Seven (G7)'s economies from 1990 to 2020 and found that renewable energy promotion has an inhibitory effect on the economic development of the region in which the economy is located.

In terms of empirical evidence for the neutrality hypothesis, Apergis [14] used panel data for 27 European countries between 1997 and 2007 to find a weak relationship between economic growth and renewable energy consumption in Europe. Liu and Hao [15] investigated the relationship between renewable energy development and GDP growth in 17 developing countries, but found no clear evidence to support the relationship in 16 countries, with the exception Poland.

The existing literature explores the relationship between renewable energy development and economic development via the application of different methods in different countries and regions. The countries and regions studied primarily include China [16], Nigeria [13], Vietnam [7], Pakistan [17], Morocco [18], developing economies [19], European countries [20], Eurasia [14], the G7 [13], Next Eleven (N-11) [8], and OECD countries [10]. We believe that the fundamental reason for the different conclusions of the existing studies is that these literatures are based on local samples, and different countries and regions have their own special endowment structures. Few studies have divided countries and regions with different endowment conditions on a global scale to provide empirical evidence of the relationship between renewable energy development and economic development, and, therefore, establish general rules. This paper complements previous work by using 31 years of global panel data for 125 countries as research samples. The contribution of this paper is as follows: (1) The dynamic evolution law of renewable energy development

to economic development is analyzed, and the general relationship between renewable energy development and economic development is discussed under different endowment conditions. (2) The majority of the existing literature on the relationship between renewable energy development and economic development only contains thought design or empirical results, and there is a lack of research in the literature to explain the mechanism of action. In this paper, the interaction term is introduced to discuss the role of the energy trade structure. (3) This study subdivides countries in the world according to their economic conditions and natural resource abundance, providing differentiated and targeted policy recommendations. At a time when the development of renewable energy has become a general trend, this study can provide general rules on how and whether countries should develop renewable energy and contributes to the limited literature on the relationship between renewable energy and economic development worldwide.

### 2.2. Research Hypothesis

#### 2.2.1. Renewable Energy Phase Characteristics Are a Contribution to the Limited Literature

In the early stage of the development of renewable energy, the lack of technological innovation comprises the primary issue that must be solved. Technology immaturity increases the cost of renewable energy generation [21]. A high cost of production sets a high price, and renewable energy lacks the ability to compete with traditional energy in the market. In this period, renewable energy is in urgent need of a large quantities of capital, technology, and talent support, and government subsidies and support remain indispensable. The capital demand for renewable energy in this period is huge and induces a crowding-out effect, negatively affecting economic development. In the mature stage of renewable energy development, the related systems and mechanisms of renewable energy are relatively perfect, and the government support policies are gradually withdrawn. In the later stage of development, the renewable energy technology becomes mature, the marginal cost of renewable energy decreases, and the price of renewable energy decreases, thus, allowing competition with traditional energy in the market. Through the scale effect and learning effect, the renewable energy industry will develop in terms of operational scale, improve the efficiency of the renewable energy industry, and gradually increase its share in the energy structure [22].

#### 2.2.2. Structure of Factor Endowment

The current level of economic development figures significantly in this discussion. According to the existing research, the level of urbanization has a great impact on the development of renewable energy [23]. With the development of urbanization, the total demand for energy consumption increases. Renewable energy sources, as complements to traditional energy sources, also grow as demand increases. The urbanization rate is generally linked to the level of economic development and, on the whole, the urbanization rate has a positive correlation with the degree of economic development [24]. Technological progress plays an important role in promoting the development of renewable energy. Stable and sufficient spending on research and development of renewable energy are important guarantees of the long-term development of renewable energy. Compared with developing countries, developed countries have higher levels of scientific and technological development, as well as personnel with a higher level of skill in navigating technology. In addition, the degree of policy support for renewable energy development varies among countries with different levels of development. Compared with developing countries, developed countries have a more rational economic and industrial structure, pursue higher-quality economic development, formulate more policies to encourage the development of renewable energy, and provide sufficient funds and a sound development environment for the primary stage of renewable energy.

Natural abundance can factor into these considerations. Different countries and regions have different resource endowments. Regional differences in the abundance and distribution of water resources, wind energy resources, and solar energy resources leads to

heterogeneity of renewable energy production and utilization in different countries and regions [25]. In countries with high natural abundance, the cost of primary products is low, the profits from resource-related industries are high, and primary industry dominates. Capital flows to primary industries squeeze out other sectors, making it difficult to provide the large amounts of capital necessary for the initial development of renewable energy. At the same time, it is difficult to achieve long-term development because of the lack of manufacturing power developed while obtaining short-term income from natural resources. The structuralist school believes that resource-based countries with a high proportion of primary industries also have a high proportion of primary products in their export products. A rapid influx of income from resource exports leads to the appreciation of the domestic currency, deteriorates the terms of trade, and makes non-resource-based industries weak in terms of competitiveness and dependent on protection and subsidies. The development of renewable energy is highly dependent on industrial policies and unstable, indirectly inhibiting the development of local renewable energy and economic growth. In addition, wealth is often more easily acquired in countries that are rich in natural resources than in countries without. This leads to resource-rich societies having a reduced incentive to invest in human capital and innovation, resulting in less investment in education and a shortage of highly skilled workers. Without the support of science and technology, there is no soil for the development of renewable energy, and there is no impetus for economic development to push innovation.

The impact of renewable energy on economic development is phased. The development of renewable energy has different effects on the economy at different stages. In the initial stage, it may require a lot of investment and government support. However, with the maturity of technology and the expansion of scale, the renewable energy industry is expected to become the engine of economic growth. The negative impact of renewable energy development on economic development may gradually weaken. Endowment conditions affect the resource availability, technological development, and market demand of renewable energy, thus, determining the scale and speed of renewable energy. Based on this, we put forward these hypotheses:

**Hypothesis 1.** *In different stages of the development of renewable energy, the intensity of impact of the development of renewable energy on economic development is different.*

**Hypothesis 2.** *Under different economic development levels, natural resource abundance, and other endowment conditions, the impact of renewable energy development on economic development is different.*

### 3. Methodology and Data

*3.1. Model Specification*

In order to explore the existence of the renewable energy curse, this paper constructs a two-way fixed-effects model of time and country. The specific model settings are as follows:

$$EG_{i,t} = \alpha + \beta RED_{i,t} + \gamma X_{i,t} + \mu_i + \lambda_t + \varepsilon_{i,t} \tag{1}$$

In this model, $i = 1, 2 \ldots, n$ denotes different countries; $t = 1, 2 \ldots, T$ represents different years; $EG_{i,t}$ represents the level of economic growth; $\alpha$ represents the constant term; $RED_{i,t}$ represents the development of renewable energy; $\gamma$ represents the coefficient of the control variables; $X_{i,t}$ represents some control variables at the national level; $\mu_i$ represents country fixed effect; and $\lambda_t$ represents time-fixed effect. $\varepsilon_{i,t}$ is a number of random perturbations. In Equation (1), coefficient $\beta$ is the key coefficient. If the coefficient $\beta$ is less than 0, renewable energy development slows economic growth.

Traditional OLS estimates are statistical methods used to estimate the relationship between independent variables and dependent variables in a linear regression model, and can also be used to determine the optimal fitting line by minimizing the sum of residual squares. Traditional OLS estimates only show the average impact of renewable energy on

economic development. To further study the impact of renewable energy on economic growth at different stages of development, we adopted a panel quantile regression method. This method estimates the influence of the independent variable on the conditional decimal point of the dependent variable based on the minimal weighted sum of the absolute residual value. Using a quantile regression model, we can analyze the influence mechanism of renewable energy development on economic growth in different development stages, and perform a more comprehensive analysis based on the ordinary least-squares method. The basic form of quantile regression model is:

$$Q_{EG_{i,t}}^{(\tau)}(\tau \mid RED_{i,t}) = \alpha + \theta(\tau)RED_{i,t} + \gamma X_{i,t} + \mu_i + \lambda_t + \varepsilon_{i,t} \tag{2}$$

In Formula (2), $\tau$ is the corresponding quantile; $Q_{EG_{i,t}}^{(\tau)}$) is the level of economic growth in the corresponding quantile; $\alpha$ represents the a constant term; $RED_{i,t}$ is the development of renewable energy under the corresponding quantile; $\gamma$ represents the coefficient of the control variables; $X_{i,t}$ represents some control variables at the national level; $\mu_i$ represents national fixed effect; $\lambda_t$ represents time fixed effect; and $\varepsilon_{i,t}$ is a number of random perturbations.

*3.2. Data and Variables*

Given the background and data integrity of renewable energy policy implementation, a 31 year global panel database is established for 125 countries. The database used in this paper is mainly composed of indicators such as renewable energy development and economic development, and the data come from the World Development Indicators (WDI) (https://datatopics.worldbank.org/world-development-indicators/, accessed on 3 January 2023.) database. In this paper, panel data from 1990 to 2022 are selected, and 2021, with serious data omissions missing, is deleted. Additionally, a small amount of missing data are completed by linear interpolation method. At the same time, 1% tail reduction was carried out on the data to reduce the possibility of estimation deviation due to extreme values. Table 1 provides detailed information on variable definitions and measurement methods.

**Table 1.** Variable definition and measures.

| Variable | Definitions and Measures |
|---|---|
| RED | Renewable energy generation, measured by the total renewable energy generation in a country |
| EG | Economic growth, which is the logarithm of a country's total GDP |
| ES | Energy–resource structure, which is the share of renewable energy consumption in the total energy consumption |
| NRA | Natural resource abundance, the logarithm of the proportion of total natural resource rent in GDP |
| TS | Technical structure, which is the value added by high-tech manufacturing industries |
| Urban | Urbanization level, which is the proportion of urban population to total population |
| Technology | Number of articles in scientific journals, which is the logarithm of the number of scientific journal papers |
| IAR | Internet access rate, the fixed broadband subscriptions (per 100 people) |
| FDI | Foreign direct investment, measured by the FDI in the proportion of GDP |
| Industrial | Industrial structure, which is the proportion of manufacturing value added in GDP |

1. Explained variable: economic growth (EG) is defined by GDP. We select the size of a country's economy as a variable with which to describe economic development. GDP directly reflects the size of a country's economy. Therefore, we take the logarithm of GDP expressed in US dollars as a measure of the size of a country's economy, and, thus, a representation of the economic development of a country;

2. Explanatory variables: renewable energy development (RED) is represented by renewable energy generation. The contribution of renewable energy to the total supply of primary energy is defined as renewable energy. Renewable energy consists of the primary equivalent/biofuel of hydro/geothermal/solar/wind/tidal/wave energy. Units may be expressed in kilotons (tons of oil equivalent) or in percentage of primary energy supply. The

power generation of renewable energy is closely related to the development of renewable energy. The power generation of renewable energy in a country can reflect the total amount of renewable energy and the national capacity to produce renewable energy. We believe that the more renewable energy a country generates, the higher its level of renewable energy development is;

3. Other control variables. In order to increase the reliability of the model, eight control variables are introduced as follows:

(1) Energy–resource structure (ES) is defined by the proportion of renewable energy consumption to total energy consumption. A rational energy structure can reduce large-scale waste and inefficient use of energy, adjust input–output imbalance, and have a positive impact on economic growth;

(2) Natural resource abundance (NRA) is represented by the share of total natural resource rent in GDP. Economic development is affected by regional resource abundance. This inverse correlation between the abundance of natural resources and economic growth is defined as the "resource curse". Total rents for natural resources are the sum of rents derived for oil, gas, coal (hard and soft coal), minerals, and forests;

(3) Technical structure (TS) is represented by medium- and high-tech manufacturing value added. Technological structure refers to the combination and proportion of different levels and types of material forms and knowledge forms of a country, department, region, or enterprise in a certain period of time, and this affects the national industrial structure and level of economic development. Medium- and high-tech manufacturing value added indicates that the country's technological level has improved;

(4) Urbanization level (urban) is represented by the proportion of urban population to the total population. Urbanization is the result of a growing population and economy under the functions of agglomeration economies and scale economies. Economic growth inevitably improves the level of urbanization, and the improvement of urbanization level also promotes economic growth;

(5) Technical level is defined by the number of articles in scientific journals published by authors from a specific place. The number of articles in scientific journals are treated with logarithms. The improvement of technological level is conducive to innovation-driven economic development. The logarithm of the number of articles published in scientific journals of a country is used to measure the total scientific and technological achievements of the country to further obtain the scientific and technological level of the country;

(6) Internet access rate (IAR) is defined by the number of fixed broadband subscriptions per 100 people. Infrastructure is the basic condition of economic development and economic modernization and affects the level and speed of economic development;

(7) Foreign direct investment (FDI) is represented by the share of FDI in GDP. Foreign direct investment in a country can enable the country to obtain external resources, improve the technical level, and organizational efficiency of the host nation through technological spillover, and, thus, accelerate the process of economic development of the country;

(8) Industrial structure (industrial) is represented by share of value added by manufacturing industry to GDP. A reasonable industrial structure is beneficial for optimizing the allocation of resources and promoting the improvement of labor productivity. An industrial structure corresponding to a country's economic level is the premise upon which healthy and stable economic development rests.

## 4. Empirical Result and Analysis

First, we performed OLS regressions on renewable energy development and economic development based on the global panel database for 125 countries from 1990 to 2021. Then, the panel quantile model was used to identify the evolutionary trajectory of the marginal effect of economic growth under different development stages of renewable energy. Subsequently, we grouped the regression into different stages of renewable energy development. Finally, we used the shock IV method, an instrumental variable method, and used exogenous shocks as instrumental variables in order to estimate causal effects when

dealing with endogenous problems [26], replace variables, and control macro factors to conduct robustness testing. Finally, we conducted grouping regression and reported the results in this paper, and the interaction term was introduced to discuss the mechanism of renewable energy development affecting economic development.

### 4.1. The Average Economic Impact of Renewable Energy

Column (1) of the Table 2 only discusses the relationship between renewable energy development and economic development. In addition to the explanatory variables and the explained variables, a series of control variables are added in column (2). On the basis of column (2), column (3) adds the time control effect, and column (4) adds the two-way fixed effect of time and region. OLS regression and two-way fixed effect show that there is a significant negative correlation between the degree of renewable energy development and economic growth. OLS and bidirectional fixed effects show that there is a significant negative correlation between the degree of renewable energy development and economic growth. Column (4) shows that the average impact of renewable energy development on economic development is −0.0130, which is significant at a 1% level. This result means that a 1% increase in the renewable energy development composite index reduces the economic development composite index by 0.013%. This indicates that, in the process of economic development, the development of renewable energy reduces the growth rate of economic development, and the curse of renewable energy does exist.

**Table 2.** OLS regression estimation results.

| Variables | EG | | | |
|---|---|---|---|---|
| | (1) | (2) | (3) | (4) |
| RED | −0.0323 *** | −0.0167 *** | −0.0136 *** | −0.0130 *** |
| | (0.0063) | (0.0035) | (0.0022) | (0.0025) |
| NRA | | 0.8509 * | −0.1955 | −0.2210 |
| | | (0.4851) | (0.5116) | (0.5403) |
| TS | | 0.0067 * | 0.0056 *** | 0.0026 |
| | | (0.0040) | (0.0021) | (0.0022) |
| Urban | | 3.3996 *** | −0.4030 | −1.2078 |
| | | (0.7209) | (0.5290) | (0.7316) |
| Technology | | 0.2936 *** | 0.0866 *** | 0.0585 ** |
| | | (0.0495) | (0.0272) | (0.0253) |
| IAR | | 1.8912 *** | −1.1380 *** | −1.3020 *** |
| | | (0.2657) | (0.2186) | (0.2401) |
| FDI | | 0.7274 ** | −0.0192 | −0.0421 |
| | | (0.3501) | (0.1502) | (0.1442) |
| Industrial | | −1.2036 | 0.1651 | 0.1749 |
| | | (0.8445) | (0.4601) | (0.4588) |
| _cons | 25.5666 *** | 20.9129 *** | 23.7597 *** | 24.4212 *** |
| | (0.2424) | (0.3947) | (0.3529) | (0.4327) |
| YearFE | No | No | Yes | Yes |
| CourtyFE | No | No | No | Yes |
| N | 3875 | 3875 | 3875 | 3875 |
| r2_w | 0.0706 | 0.6221 | 0.8342 | 0.8367 |

Note: Numbers in brackets are standard errors. The asterisk indicates the significance of the corresponding level, *** (1%), ** (5%), * (10%).

### 4.2. The Impact of Renewable Energy on Economic Growth at Different Stages of Development

The results given by the above grouping regression are only average influences, which cannot identify the evolution track of the marginal effects of economic growth under different development stages of renewable energy, let alone determine whether the structural change point of the result exists. The panel quantile regression technique is used to estimate the econometric model. We take the renewable energy generation as the evaluation criterion for the maturity of a country's renewable energy development, divide

the development of renewable energy into several sections, and obtain three sample groups: the early-stage sample group, the growth-stage sample group, and the mature-stage sample group. These constitute the samples where the proportion of renewable energy in the energy mix is in the bottom 30%, 30–70%, and 70–100%, respectively.

The Table 3 regression results show that the estimated coefficient of RED is negative, but that it does not pass the significance test. Renewable energy has no significant impact on the economy in the early stage of development. In the ranges of 30–70% and 70–100% of the development of renewable energy, the estimated coefficient of RED is negative and significant at the level of 1%. This indicates that when the development of renewable energy enters the growth and maturity stage, the development of renewable energy exerts a more and more obvious inhibitory effect on economic growth, resulting in the curse of renewable energy. This indicates that in the early stage of the development of renewable energy, renewable energy does not require much capital and technology, and does not affect economic development. However, when renewable energy enters the growth and maturity stage, the high cost of research and development brings obvious substitution and crowding out effect to the economy, which has a negative impact on the economic development.

**Table 3.** Grouped regression estimates results.

| Variables | EG | | | | | |
|---|---|---|---|---|---|---|
| | (1) | (2) | (3) | (4) | (5) | (6) |
| RED | −0.0091 | −0.0278 | −0.0132 *** | −0.0141 *** | −0.0167 *** | −0.0158 *** |
| | (0.0274) | (0.0274) | (0.0042) | (0.0043) | (0.0029) | (0.0032) |
| NRA | −0.3433 | −0.5726 | −3.9305 *** | −4.0928 *** | 0.3535 | 0.2972 |
| | (0.4336) | (0.4437) | (0.9837) | (0.8452) | (0.6239) | (0.6525) |
| TS | 0.0060 | 0.0012 | 0.0012 | −0.0025 | 0.0076 | 0.0027 |
| | (0.0038) | (0.0038) | (0.0036) | (0.0036) | (0.0047) | (0.0045) |
| Urban | −2.0411 ** | −4.3802 *** | 0.1428 | −0.7975 | 0.9191 | 0.7006 |
| | (0.9936) | (1.5497) | (0.8038) | (1.2648) | (0.5967) | (0.8455) |
| Technology | 0.2301 *** | 0.1563 *** | 0.0667 ** | 0.0307 | 0.0470 | 0.0072 |
| | (0.0432) | (0.0455) | (0.0278) | (0.0189) | (0.0341) | (0.0366) |
| IAR | −1.3578 *** | −1.5866 *** | −0.7274 ** | −0.8951 ** | −0.7613 ** | −0.8995 ** |
| | (0.4849) | (0.5183) | (0.3554) | (0.3732) | (0.3463) | (0.3713) |
| FDI | −0.0974 | 0.0458 | 0.0417 | −0.0575 | −1.0577 ** | −0.9825 ** |
| | (0.1695) | (0.2065) | (0.2186) | (0.2189) | (0.4235) | (0.4533) |
| Industrial | −0.1957 | 0.1545 | 1.0656 | 0.8082 | 0.0734 | −0.1082 |
| | (0.6390) | (0.6364) | (1.0633) | (0.9647) | (1.0780) | (1.1254) |
| _cons | 24.0757 *** | 26.3141 *** | 23.8016 *** | 24.8313 *** | 23.2462 *** | 23.5082 *** |
| | (0.8469) | (1.1531) | (0.5520) | (0.7590) | (0.3457) | (0.3557) |
| YearFE | Yes | Yes | Yes | Yes | Yes | Yes |
| CourtyFE | No | Yes | No | Yes | No | Yes |
| N | 1162 | 1162 | 1550 | 1550 | 1163 | 1163 |
| r2_w | 0.7983 | 0.8098 | 0.8537 | 0.8592 | 0.8839 | 0.8861 |

Note: Numbers in brackets are standard errors. The asterisk indicates the significance of the corresponding level, *** (1%), ** (5%).

### 4.3. Heterogeneity Analysis

This paper uses the panel quantile regression model to observe and analyze the dynamic evolution trajectory of the renewable energy curse under different endowment conditions. Considering all control variables at the same time, the evolution trend of the renewable energy curse is revealed. Figure 2 shows the marginal evolution trend of economic growth in relation to the different stages of renewable energy development in countries with varied development levels.

In developed countries, when the economic development level is from 0% to 80%, the marginal effect of renewable energy development on economic growth is significantly negative and tends to decline with economic development. However, when the quantile of economic development is greater than 80%, the marginal utility of renewable energy devel-

opment on economic growth becomes significantly negative, and the absolute marginal utility decreases rapidly. This suggests that the curse of renewables is growing in the rich world as economies grow. However, when the economy develops to a certain level, the curse effect gradually wanes.

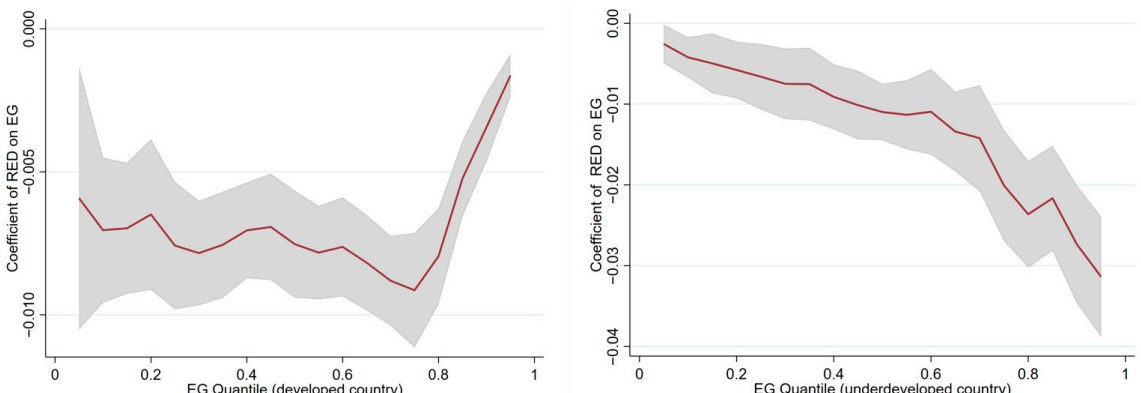

**Figure 2.** Marginal effects of renewable energy development on countries with different economic development levels.

In developing countries, when the economic development level is 0% to 60%, the marginal utility of renewable energy development on economic growth is significantly negative and tends to decline with the development of the economy. When the quantile of economic development is greater than 60%, the marginal utility of renewable energy development on economic growth is significantly negative, and the absolute marginal utility increases sharply. This means that in developing countries, the curse of renewable energy increases gradually with economic development and that, after a certain level of economic development, the curse effect becomes dramatic.

Figure 3 shows the marginal evolution trend of the renewable energy curse under different endowment conditions. In resource-abundant countries, the impact of renewable energy development on economic growth is initially negative and diminishes as the economy progresses up to the 80% mark outlined. However, when economic development progresses beyond 80%, the negative marginal utility of renewable energy on economic growth increases rapidly, indicating an intensification of the curse of renewable energy in these countries. Conversely, in resource-poor countries, the negative marginal utility of renewable energy on economic growth strengthens and declines at a faster rate as the economy develops. This suggests a more pronounced curse of renewable energy in resource-poor countries as their economies advance.

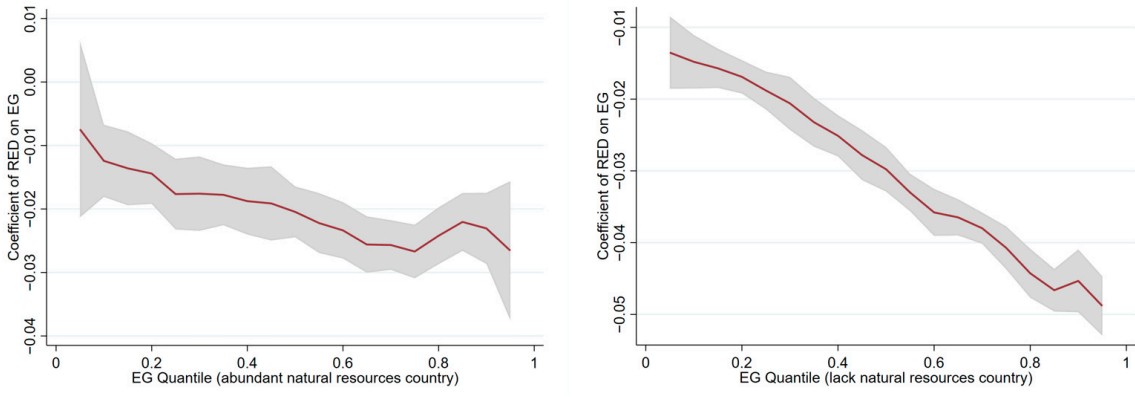

**Figure 3.** Marginal effects of renewable energy development on countries with different natural resource abundance.

These results suggest that, at a certain point, a renewable energy curse will emerge fully. However, the curse of renewable resources varies among countries at different levels of development. In Table 4, columns (1) and (3) only control the fixed effect at the time level, whiles column (2) and (4) realize the bidirectional fixed effect between time and country. Columns (1)—(2) report the results of the sample from developed countries, and columns (3)—(4) report the results of the developing country sample. The absolute value of RED coefficient in developed countries is significantly smaller than that in underdeveloped countries. This is yet another example of the renewables curse and suggests that the use of renewables in developed countries constitutes less of a drag on the economy than it does in developing countries.

**Table 4.** Regression estimates by regional level of development.

| Variables | (1) | (2) | (3) | (4) |
|---|---|---|---|---|
| | \multicolumn EG | | | |
| | Developed | | Underdeveloped | |
| RED | −0.0102 *** | −0.0107 *** | −0.0206 *** | −0.0255 *** |
| | (0.0023) | (0.0028) | (0.0038) | (0.0048) |
| NRA | 0.5136 | 0.4830 | −1.2147 | −1.1046 |
| | (0.4300) | (0.4835) | (0.8780) | (0.8879) |
| TS | 0.0002 | −0.0030 | 0.0019 | 0.0007 |
| | (0.0026) | (0.0027) | (0.0024) | (0.0023) |
| Urban | 0.0029 | −0.7808 | −0.2990 | −0.7932 |
| | (0.5023) | (0.8425) | (0.5593) | (0.9540) |
| Technology | 0.0323 | 0.0090 | 0.1502 ** | 0.0950 |
| | (0.0204) | (0.0192) | (0.0670) | (0.0642) |
| IAR | −0.3315 | −0.3765 | −0.5822 | −0.7171 * |
| | (0.2923) | (0.3098) | (0.3819) | (0.4268) |
| FDI | −0.0594 | −0.0743 | 0.0142 | 0.0510 |
| | (0.1545) | (0.1577) | (0.0964) | (0.1009) |
| Industrial | 0.3707 | 0.2997 | 0.6358 | 0.7745 |
| | (0.5573) | (0.5689) | (0.7328) | (0.7150) |
| _cons | 22.9473 *** | 23.1706 *** | 24.7145 *** | 25.8310 *** |
| | (0.3220) | (0.4390) | (0.6332) | (0.8290) |
| YearFE | Yes | Yes | Yes | Yes |
| CourtyFE | No | Yes | No | Yes |
| N | 2223 | 2223 | 1652 | 1652 |
| r2_w | 0.8212 | 0.8236 | 0.8424 | 0.8459 |

Note: Numbers in brackets are standard errors. The asterisk indicates the significance of the corresponding level, *** (1%), ** (5%), * (10%).

Natural resources have a strong geographical nature. Renewable energy sources such as solar energy, wind energy, and hydropower depend on the existence and utilization of natural resources. Therefore, the intensity of the renewable energy curse has the same heterogeneity of resource endowment. The initial static cost of renewable energy comes from regional natural resource endowment conditions [27]. The different geographical locations, climatic conditions, and natural reserves of a country or region also affect the price and cost of renewable energy in different regions, as well as economic benefits of their use. In Table 5, columns (1) and (3) only control the fixed effect at the time level, while columns (2) and (4) realize the bidirectional fixed effect between time and country. Columns (1)–(2) report the results of samples from countries with high resource abundance, and columns (3)–(4) report the results of samples with low resource abundance. By comparing the absolute value of the coefficient, we can determine that the renewables curse is stronger in countries with abundant natural resources. This suggests that the curse of renewable energy exists regardless of the level of resource abundance. Countries with high natural abundance suffer more from the curse of renewable energy than those with low natural abundance. A possible reason for this is that resource-rich countries often suffer

from the "resource curse". In addition, economically developed regions with relatively developed manufacturing industries gradually transfer energy-intensive industries to these resource-rich regions, hindering the low-carbon transformation of the local energy structure. The lack of motivation and technical level for the development of renewable energy makes it difficult to meet the high requirements for technology and capital in the development process of renewable energy. Therefore, these factors magnify the negative impact of the development of renewable energy on the economy.

**Table 5.** Regression estimates by regional natural resource abundance.

| Variables | (1) | (2) | (3) | (4) |
|---|---|---|---|---|
| | EG | | | |
| | Abundant | | Shortage | |
| RED | −0.0024 | −0.0359 *** | −0.0134 *** | −0.0129 *** |
| | (0.0047) | (0.0118) | (0.0022) | (0.0025) |
| NRA | 0.0914 | −0.5834 | −2.0947 *** | −2.1888 *** |
| | (0.9356) | (0.7204) | (0.8034) | (0.7956) |
| TS | 0.0239 *** | −0.0022 | 0.0040* | 0.0012 |
| | (0.0078) | (0.0066) | (0.0022) | (0.0021) |
| Urban | 0.5958 | −5.7508 | −0.5398 | −1.2256 * |
| | (0.5275) | (3.5109) | (0.5214) | (0.7032) |
| Technology | 0.3606 *** | 0.1629 ** | 0.0565 *** | 0.0330 * |
| | (0.0796) | (0.0598) | (0.0211) | (0.0186) |
| IAR | −0.1275 | 1.1335 | −0.9701 *** | −1.0820 *** |
| | (2.0873) | (2.0479) | (0.2087) | (0.2190) |
| FDI | −2.6135 *** | −0.5190 | 0.0887 | 0.0740 |
| | (0.7755) | (0.6367) | (0.1217) | (0.1190) |
| Industrial | −0.5575 | −1.8137 * | 0.8013 | 0.8451 |
| | (1.0907) | (1.0506) | (0.5588) | (0.5370) |
| _cons | 21.1830 *** | 27.2495 *** | 24.0282 *** | 24.6208 *** |
| | (0.7580) | (2.2973) | (0.3625) | (0.4265) |
| YearFE | Yes | Yes | Yes | Yes |
| CourtyFE | No | Yes | No | Yes |
| N | 511 | 511 | 3364 | 3364 |
| r2_w | 0.7087 | 0.8391 | 0.8691 | 0.8712 |

Note: Numbers in brackets are standard errors. The asterisk indicates the significance of the corresponding level, *** (1%), ** (5%), * (10%).

### 4.4. Robustness Test

Table 6 reports the robustness test results. Column (1) and column (2) use the shock IV method and uses the shock of fixed feed-in tariff policy as an instrumental variable for the development of renewable energy.

In order to further eliminate the impact of measurement errors of renewable energy development indicators on the conclusion, core explanatory variables are replaced in column (3)–column (4). In column (3), the renewable energy generation used above is replaced by the proportion of renewable energy generation in the total energy generation, and the estimated coefficient of RED is significantly negative at the level of 1%. This indicates that the greater the proportion of clean energy in the total energy output there is, the slower economic development will be. In column (4), we see that the energy consumption structure is used to replace renewable energy generation. The estimated coefficient of ES is significantly negative. The greater the proportion of clean energy in the whole energy structure is, the stronger the inhibition effect on economic development is. The original conclusion is robust.

There may be some common factors affecting the development conditions of renewable energy in the same region that are not included in the regression equation. We processed column (5) by excluding the influence of macroeconomic factors. It reduces the possibility that some ignored inherent conditions interfere with the regression results. After testing,

the estimated coefficient of RED is still significantly negative at the level of 1%. This allows us to conclude that our conclusions are still reliable under the condition that macro factors are taken into account.

**Table 6.** Empirical results of robustness test.

| Variables | (1) | (2) | (3) | (4) | (5) |
|---|---|---|---|---|---|
| | | | EG | | |
| | Shock IV | | Proportion of Renewable Energy Generation | Energy Consumption Structure | Excluding macro Factors Affect |
| RED | −0.0210 *** | −0.0142 *** | | | −0.0135 *** |
| | (0.0015) | (0.0010) | | | (0.0025) |
| NRA | 0.8713 *** | −0.1082 | −0.3107 | −0.2746 | −0.2594 |
| | (0.1561) | (0.1109) | (0.5210) | (0.5290) | (0.5568) |
| TS | 0.0092 *** | 0.0038 *** | 0.0021 | 0.0022 | 0.0024 |
| | (0.0014) | (0.0009) | (0.0022) | (0.0022) | (0.0023) |
| Urban | 5.0996 *** | −0.7680 *** | −0.5870 | −1.0789 | −1.2515 * |
| | (0.2524) | (0.1978) | (0.7412) | (0.7193) | (0.7514) |
| Technology | 0.2223 *** | 0.0502 *** | 0.0625 ** | 0.0652 ** | 0.0553 ** |
| | (0.0095) | (0.0070) | (0.0258) | (0.0267) | (0.0257) |
| IAR | 1.7481 *** | −1.2412 *** | −1.6197 *** | −1.3098 *** | −1.3300 *** |
| | (0.0927) | (0.0791) | (0.2346) | (0.2518) | (0.2485) |
| FDI | 0.8112 *** | −0.0538 | −0.0281 | −0.1674 | −0.0626 |
| | (0.1220) | (0.0841) | (0.1447) | (0.1701) | (0.1627) |
| Industrial | −1.7424 *** | 0.0452 | 0.2783 | 0.3735 | 0.1581 |
| | (0.2035) | (0.1399) | (0.5175) | (0.4798) | (0.4608) |
| RED1 | | | −0.0083 * | | |
| | | | (0.0047) | | |
| ES | | | | −0.0094 *** | |
| | | | | (0.0032) | |
| _cons | | | 23.7208 *** | 24.1995 *** | 24.5317 *** |
| | | | (0.4473) | (0.4361) | (0.4419) |
| YearFE | No | Yes | Yes | Yes | Yes |
| CourtyFE | Yes | Yes | Yes | Yes | Yes |
| N | 3410 | 3410 | 3875 | 3875 | 3751 |
| r2_w | | | 0.8298 | 0.8337 | 0.8342 |

Note: Numbers in brackets are standard errors. The asterisk indicates the significance of the corresponding level, *** (1%), ** (5%), * (10%).

## 5. Mechanism Analysis

The above empirical evidence proves that the development of renewable energy has an inhibitory effect on economic development. We believe that different countries or regions have different factor endowments, which are reflected in the development process of renewable energy and affect economic development through changes in trade structure.

### 5.1. The Mechanism Explanation of the Impact of Trade Structure on the Renewable Energy Curse

5.1.1. Trade Openness and Energy Development Interact

According to factor endowment theory, international trade originates from different factor stock ratios among trading countries and factor endowments. In the process of the development of renewable energy, the input of large-scale production factors, including human capital, material goods, and various intermediate inputs, boosts the output of renewable energy considerably and promotes renewable energy exports [28]. At the same time, the export of renewable energy also increases the use of corresponding raw materials and equipment, thus, increasing energy demand [29]. As renewable energy becomes an increasingly indispensable part of the energy mix, the demand for renewable energy increases accordingly and promotes the development of renewable energy [30].

### 5.1.2. Learning Effect and Scale Effect

When a country has a high degree of renewable energy development, this indicates that its renewable energy has passed the early stage of development when construction costs and technology research and development costs are extremely high. At this time, the scale effect and learning effect of renewable energy come into play, and the expansion of the scale of renewable energy causes the higher upfront costs to be shared [31]. At the same time, the continuous development of innovative renewable energy technologies reduces operation and maintenance costs. The learning effect takes effect, which not only promotes the accumulation of human capital but also enables the transmission of renewable energy technology. Both work to reduce marginal costs, which reduces the average cost of renewable energy in the long run. This allows countries with a high degree of renewable energy development to occupy a dominant position in international renewable energy transactions, creates conditions for renewable energy export, and affects the energy trade structure of a country. Both importing and exporting trade create momentum for an economy. According to the existing literature, changes in import and export trade structure have different impacts on the economy. Additionally, when the trade structure is unreasonable, this negative impacts the economy [32].

### 5.1.3. Structural Effect

The trade in renewable energy trade alters a country's energy structure. According to the existing research, the driving effect of the increase in export value of high-value-added products on a country's economy is significantly higher than that of low-value-added products [33]. Renewable energy is a high-value-added product because of its high construction cost and research cost and high technology content. The trade of renewable energy can reduce the economic problems caused by the price fluctuation of fossil energy, optimize the traditional energy structure, promote the economic transformation and upgrading of a country, and accelerate economic and social progress and the development of enterprises.

### 5.1.4. Technical Effect

Technology has strong positive externalities. In international trade, firms with advanced technologies, processes, and expertise tend to diffuse renewable energy technologies. The use of advanced technology is the key to saving costs and increasing output. In order to maximize profits, enterprises continue to strengthen research and development, cultivate scientific and technological talents, introduce advanced equipment, accelerate the development of renewable energy, and improve the production efficiency of renewable energy [34]. Overall, the aim is to constantly reduce marginal cost and increase economic benefits. In the process of technological development and innovation of enterprises, it is necessary to pay attention to adapting to the endowment structure of enterprises so as not to be counterproductive [35].

The basic reason for the inconsistencies in the existing studies is that most of them ignore the regional endowment structure of countries. This paper argues that the effects of trade structure differ under different economic development levels and levels of natural resource abundance. Based on this, this paper uses grouping test and interaction terms to establish a model to test the mechanism of renewable energy development affecting economic development. The model is set as follows:

$$EG_{i,t} = \beta RED_{i,t} + \gamma EI_{i,t} + \theta ctrl_{i,t} + prov_{i,t} + \varepsilon_{i,t} \tag{3}$$

$$EG_{i,t} = \alpha EIRED + \beta RED_{i,t} + \gamma EI_{i,t} + \theta ctrl_{i,t} + prov_{i,t} + \varepsilon_{i,t} \tag{4}$$

Formula (3) is the influence of renewable energy development on economic growth when control variables EI, RED, NRA, TS, urban, technology, IAR, and FDI are added. EI represents the percentage of a country's net energy import in the country's total energy consumption and is used to represent a country's energy trade structure. Model 4 is based on the use of model 3 to increase the cross-term between net energy imports and a country's

renewable energy generation. If the cross-term coefficient $\alpha$ is significantly greater than 0, optimizing the trade structure can significantly alleviate the negative impact of renewable energy on economic development. Irrespective, it significantly enhances the negative impact of renewable energy on economic development.

*5.2. Energy Trade Structure and Economic Development under Different Economic Development Levels*

The world sample was divided into two groups. In Table 7, column 1 and column 2 are the regression results of economically developed countries under the time fixed effect, and columns 3 and 4 are the regression results of economically developing countries under the time fixed effect. According to the mechanism test results, the estimated coefficients of cross-terms in column 1 and column 2 in Table 7 are significant at the 95% level, and the coefficients are all positive. This suggests that the larger the percentage of net energy imports is as a percentage of total energy use in developed countries, the stronger the inhibitory effect of renewable energy development on economic development are. The estimated coefficients of cross-terms in columns 3 and 4 of Table 7 are obviously negative. This suggests that the larger the percentage of net energy imports is in total energy use in developing countries, the more the inhibition effect of renewable energy development on economic development can be mitigated.

**Table 7.** Empirical results of trade structure under different economic development levels.

| Variables | EG | | | |
|---|---|---|---|---|
| | (1) | (2) | (3) | (4) |
| EIRED | 0.0022 ** | 0.0024 ** | −0.0018 ** | −0.0016 * |
| | (0.0011) | (0.0012) | (0.0008) | (0.0009) |
| EI | −0.1816 ** | −0.1883 ** | 0.0405 | 0.0457 |
| | (0.0771) | (0.0794) | (0.0513) | (0.0509) |
| RED | −0.010 7*** | −0.0125 *** | −0.0216 *** | −0.0277 *** |
| | (0.0024) | (0.0035) | (0.0049) | (0.0056) |
| NRA | −0.4773 | −0.4741 | −0.9929 | −0.9109 |
| | (0.3543) | (0.3817) | (0.6573) | (0.6858) |
| TS | 0.0018 | −0.0009 | 0.0027 | 0.0019 |
| | (0.0023) | (0.0024) | (0.0027) | (0.0027) |
| Urban | −0.3399 | −1.5208 * | −0.3333 | −0.7686 |
| | (0.4921) | (0.8773) | (0.6146) | (0.9678) |
| Technology | 0.0357 * | 0.0125 | 0.2352 *** | 0.1819 *** |
| | (0.0194) | (0.0194) | (0.0583) | (0.0582) |
| IAR | −0.3979 | −0.4551 | −0.3321 | −0.3886 |
| | (0.3488) | (0.3691) | (0.3159) | (0.3131) |
| FDI | −0.0353 | −0.0463 | 0.0655 | 0.1220 |
| | (0.1935) | (0.2013) | (0.1235) | (0.1389) |
| Industrial | 0.7668 | 0.7363 | −0.2643 | −0.1836 |
| | (0.5287) | (0.5048) | (0.9535) | (0.9095) |
| _cons | 23.0617 *** | 23.5987 *** | 24.3055 *** | 25.3732 *** |
| | (0.3098) | (0.4524) | (0.5498) | (0.7177) |
| YearFE | Yes | No | Yes | No |
| CourtyFE | Yes | Yes | No | No |
| N | 1823 | 1823 | 1272 | 1272 |
| r2_w | 0.8132 | 0.8167 | 0.8652 | 0.8679 |

Note: Numbers in brackets are standard errors. The asterisk indicates the significance of the corresponding level, *** (1%), ** (5%), * (10%).

The above results in Table 7, columns 1 to 4, show that the energy trade structure is indeed a transmission mechanism of the effect of renewable energy development on economic development. Under different economic development levels, the role of energy trade structure in the process of renewable energy development affecting economic development is different.

*5.3. Energy Trade Structure under Different Natural Resource Abundance and Economic Development*

We divided the whole sample into regions, one with a high abundance of natural resources and the other with nations with a low abundance of natural resources and conducted a regression test on the two subsamples obtained via grouping. Columns 1 and 2 show the regression results of countries with high natural resource abundance, and columns 3 and 4 are the regression results of countries with high natural resource abundance. The results of the first and second columns show that the estimation coefficient of cross-terms is significantly positive at the 95% level. This indicates that in countries with high natural source abundance, the greater the percentage of total energy consumption into pockets is, the greater the negative impact of renewable energy development on the economy is. The coefficients of three columns and four cross-terms are not significant. They show the mechanism test of the energy trade structure as the intermediate variable fails in the regions with low natural resource abundance, which indicates that the development of renewable energy cannot reduce the negative effect on economic development by adjusting the energy trade structure in such regions.

## 6. Conclusions

This work studied the relationship between renewable energy development and economic development. OLS, quantile regression, group regression, shock IV, and other methods are used to estimate the data. Empirical analysis shows that there is a renewable energy curse in the world, and that the development of renewable energy comes at the expense of certain economic benefits. Specifically, the renewable energy curse is gradually emerging with its own development; the renewable energy curse of developed countries has weakened with economic development, and the renewable energy curse of developing countries has continued to increase with economic development. The economic development of countries with scarce natural resources is more vulnerable to the impact of renewable energy development, and the change in trade structure affects the specific role between the two.

Through the study of 125 countries around the world, this study established the general law between renewable energy development and economic development. At the moment of low-carbon development and energy structure adjustment, this is of great significance to ensuring high-quality economic development. First, policy makers and relevant stakeholders should face up to the negative effects of renewable energy development and formulate targeted policies to promote the balance between economic growth and sustainable development. Secondly, in different stages of renewable energy development, we should distinguish between development strategies, fully consider the impact of renewable energy development, and allocate resources in a targeted manner. Finally, it is necessary to optimize the layout and planning of renewable energy in combination with the specific national conditions of each country: countries with high levels of economic development should strengthen renewable energy development, support renewable energy technology innovation and R & D, and increase local renewable energy production; countries with low levels of economic development are seeing growth slow as they promote renewable energy development. Countries with high natural resource abundance promote renewable energy development, whereas countries with low natural resource abundance reduce renewable energy development. The research mechanism test part of this study only focuses on the changes in trade structure, and further research is needed to deepen the understanding of the mechanism by which renewable energy development impacts economic development.

**Author Contributions:** Conceptualization: Y.L.; methodology: Y.L.; data curation: H.W.; formal analysis: R.L.; writing—original draft: R.L.; writing—review and editing: R.L. and X.H.; supervision: X.H. All authors have read and agreed to the published version of the manuscript.

**Funding:** This research was funded by the 2022 National College Students' Innovation and Entrepreneurship Training Program (202210403039).

**Institutional Review Board Statement:** Not applicable.

**Informed Consent Statement:** Not applicable.

**Data Availability Statement:** The data presented in this study are available on request from the corresponding author.

**Conflicts of Interest:** The authors declare no conflict of interest.

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
