# Peer review of "Renewable Energy: A Curse or Blessing—International Evidence"

_sustainability, doi:10.3390/su151411103_

Round 1

Reviewer 1 Report

This paper used a panel of 125 countries in the world from 1990 to 2021 to empirically test the impact of renewable energy on economic growth. The results show that the paper contains new and novel contributions. This manuscript has been investigated from both technical and methodological viewpoints. Kindly revise the paper based on my suggestions below.

In the Abstract, I noticed some incomplete sentences. See P1 Line 13/14. Kindly revisit this.

Under the Introductory section, I expect a summary of other sections to be included in the last para. This makes the reader to be engaged in your research.

In the data and variables section, you stated that "a 31-year global panel database for 125 countries is established". Some justifications are required to back up your sample and data. I have raised the following questions that will assist in your justification. How did you select 125 countries? Why did you exclude other countries not considered in your study? What sampling techniques was used? Why did you choose 1990 as your starting period? Are there any major events that occurred before or in 1990? 

Can you include a table to show your measurement of variables? 

Kindly include the limitation of the study and suggestions for further research in the conclusion section.

Your paper will benefit from English language editing tools like Grammarly. This will improve the quality of your discussion.

Reviewer 2 Report

This paper uses empirical data of 125 countries over period of cca 30 years to estimate impact of renewable energy on economic growth using several regression tests (i.e. quantile regression). Several country factors were taken into account, such as developed/developing countries, rich/poor natural resources, etc. The authors obtained several robust conclusions which confirm existence of conditions in which renewable energy can have negative impact on economic development. Several hypotheses were tested which lead to very general conclusions: (1) different renewable impact exists in different economic development levels; (2) different renewable development stage has different impact on economy level.

The paper is quite voluminous and presents (on basic level) methodology and data grouping which is necessary to explore possible negative effects of renewable energy expansion, taking into account different criteria. Several shortcomings were noticed in the manuscript so authors are encouraged to address them thoroughly. Revision of the English grammar is also recommended.

The following comments are related to manuscript numbers addressing line numbers in brackets.

(4) missing blank after China;

(9-22) the Abstract section should be written in concise form without conclusions using numerated structure, i.e. without (2), (3) and (4); I would suggest using such formatting in Conclusions

(19) please consider using different phrase instead awkward “embryonic” since it’s not something typical to be used, consider using “early stage” or “at the beginning” etc. Similar comment goes to “crowd out” phrase since it’s written more as a slang term, consider using alterative. This should be changed on all instances in text.

(27) missing reference to 2022 COP27

(29) please expand all acronyms at the first occurrence, missing it at UNEP -> this should be harmonized throughout the text

(76) missing capital letter at Figure 1

(82) reference [6] has a different placing i.e. it uses a blank space in front -> this is not harmonized in text so you should decided and keep only one formatting (you have this on many places in text)

(92) “Who” should be “who”

(95) missing blank space after reference [8], you should decided and keep only one formatting

(101)  A. Leonard is reference [12], not [11]

 (119) missing “and” before OECD

(122) consider alternative to “research stations”, is sounds awkward

(123) strange formatting error “pro-vide”; you have this in many places (134, 142, 145, 151, 152, 153, …) please correct this

(133) “Subdivide countries” should probably be “Subdivision of countries”…. “and providing…”

(140) missing capital “R” in 2.2.

(147-148) consider “The massive capital demand of renewable …”; keep it simple and short and try not to repeat your sentences

(187) “currency appreciate” sounds awkward, please rephrase it

(190) missing sentence continuation, i.e. “… unstable, indirectly …”

(198) you should probably give one sentence of motivation before introducing hypotheses; written like this it’s not connected to previous text

(209) EGi,t should be without underscore; missing what are coefficients α and γ

(214) what is OLS?

(222) equation 2 now has variables without commas in their indexes like in equation 1; please harmonize it in text

(232) expand acronym WDI and add reference since you are using this database

(258) expand acronym TNR

(288) missing capital “I” and one bracket is extra

(299) what is Shock-IV method? Please provide reference to it

(304) “table” should be “table 1”; all tables in text have typo “Indutrial” and different labels like “_cons” or “constant” -> decide which one to use

(315) last four rows in table 1 are not mentioned in text, they should be explained or deleted

(325) consider something else instead inappropriate “germination stage” phrase

(328) you are repeating similar unclear phrase on all instances when discussing tables, “coefficient before column (1) – (2) RED” -> you have to rewrite this since you are confusing the reader, state it clearly and simple, this is very unclear

(330) please rephrase inappropriate “embryonic” on all instances

(331) what is “coefficient before RED” ???

(345-348) this sentences don’t makes sense, it’s probably copy-paste error, please rewrite

(355) should “negative” be “positive” since there is positive gradient in curve? Please check it and consider rewriting this section

(367) typo “.” In Figure 2 caption

(369-382) you are repeating the same sentences with trivial modifications, all this can be said in just several sentences, please keep it short and simple since you will lost reader’s interest

(391) again “coefficient before RED”; state it clearly

(399) consider rewriting this sentence

(408) same like in (391); columns (1) should probably be column (1)

(423) wrong caption number -> Table 6 should be 4; again in (443)

(427-428) “Table(3) – Table(4)” should be columns as I understand

(430, 433, 440) see (391) please rephrase it to be clear

(443) regarding table -> “marco facors” should be “macro factors”

(456) “material materials” should be rephrased

(497) sentence is unclear

(510-517) repeating similar things, keep it short

(520) Table 8 should be Table 5; columns (12)? Table caption referencing is wrong on all other instances

(537) should be before table since it’s a title

(538-539) should be after the table

(544) what is “table of wines” ???

(556) and (557-558) same comments as above -> table title ; why italic variables in the last tabe?

(559) Conclusions section need to be rewritten, it can’t be 2 full pages long; section (581-617) should be deleted since it’s copy-paste of the working text

Revision of the English grammar is recommended

Round 2

Reviewer 2 Report

The revised paper is greatly improved and more thoroughly explains the impact of renewable energy on economic growth using several regression tests, which are based on empirical data of 125 countries, covering last 30 years. The presented analysis in the manuscript has a more logical and complete way than the original version of the manuscript. 

The authors have also addressed all of the raised questions in a satisfactory way, so I would recommend this revised version to be accepted by the Journal in the present form.